# Development of the Integrated Glaucoma Risk Index

**DOI:** 10.3390/diagnostics12030734

**Published:** 2022-03-17

**Authors:** Sejong Oh, Kyong Jin Cho, Seong-Jae Kim

**Affiliations:** 1College of Software Convergence, Jukjeon Campus, Dankook University, Yongin 16890, Korea; sejongoh@dankook.ac.kr; 2Department of Ophthalmology, College of Medicine, Dankook University, 119, Dandae-ro, Dongnam-gu, Cheonan-si 31116, Korea; 3Department of Ophthalmology, Institute of Health Sciences, Gyeongsang National University College of Medicine and Gyeongsang National University Hospital, Jinju 52828, Korea

**Keywords:** glaucoma, machine learning, prediction, risk index

## Abstract

Various machine-learning schemes have been proposed to diagnose glaucoma. They can classify subjects into ‘normal’ or ‘glaucoma’-positive but cannot determine the severity of the latter. To complement this, researchers have proposed statistical indices for glaucoma risk. However, they are based on a single examination indicator and do not reflect the total severity of glaucoma progression. In this study, we propose an integrated glaucoma risk index (I-GRI) based on the visual field (VF) test, optical coherence tomography (OCT), and intraocular pressure (IOP) test. We extracted important features from the examination data using a machine learning scheme and integrated them into a single measure using a mathematical equation. The proposed index produces a value between 0 and 1; the higher the risk index value, the greater the risk/severity of glaucoma. In the sanity test using test cases, the I-GRI showed a balanced distribution in both glaucoma and normal cases. When we classified glaucoma and normal cases using the I-GRI, we obtained a misclassification rate of 0.07 (7%). The proposed index is useful for diagnosing glaucoma and for detecting its progression.

## 1. Introduction

Glaucoma is the leading cause of irreversible blindness around the world and progressively affects the optic nerve [1]. In Korea, the number of glaucoma patients increased from 4.01 million in 2009 to 9.79 million in 2019, with an average annual increase rate of 9.3%. The rapid increase in the number of glaucoma patients is a global trend. Therefore, accurate diagnosis of glaucoma is important. Recently, machine-learning schemes have been widely used for glaucoma diagnosis [2,3,4,5,6,7]. Classification is a major technology used in medical applications because it can be used for prediction (diagnosis). Monitoring a subject’s glaucoma potential or progression is as important as diagnosing it. Although a classification model can indicate whether the target subject is glaucoma-positive or normal, it cannot indicate how normal or severe their condition is. A few researchers have proposed the glaucoma risk index (GRI) to express the progression of glaucoma.

Bock et al. [8] presented early research results for the GRI. They performed preprocessing to eliminate disease-independent variations from the input fundus image, applied feature extraction to transform the preprocessed input data into a characteristic and compact representation, and finally built the GRI using a two-stage probabilistic support vector machine classifier. The proposed two-stage GRI was used to classify glaucoma and achieved an area under the receiver operating characteristic curve of 88% and a sensitivity of 73% at a specificity of 85%.

Loewen et al. [9] suggested a simple glaucoma index (GI) that combined preoperative intraocular pressure (IOP), number of preoperative medications, and visual field damage to capture relative glaucoma severity and resistance to treatment. The authors developed a four-level GI based on the severity level of the collected features. This GI expresses four groups rather than a range of continuous values. There was no mathematical equation for integrating the feature values. This GI was mainly used to compare the outcomes of ab interno trabeculectomy stratified by the GI.

Mookiah et al. [10] extracted various features from fundus images and built a glaucoma classification model. They also suggested a GRI using higher-order spectra and discrete wavelet transform features. The GRI values ranged between 0 and 35. In their research, the GRI was proposed as another glaucoma classifier that used a single feature.

Acharya et al. [11] extracted various features from 510 fundus images and performed principal component analysis. They constructed a GRI using five principal components (PC1–PC5). Equation (1) represents the GRI, which was obtained using a trial-and-error method. However, they did not provide a detailed analysis of their GRI, except for a simple comparison of the mean GRI values between the glaucoma and normal groups.
GRI = 6.8375 − 1.1325 × PC1 + 1.6500 × PC2 + 2.7225 × PC3+ 0.6750 × PC4 + 0.6650 × PC5(1)

Acharya et al. [12] presented a new methodology and computerized diagnostic method. They extracted pattern features from fundus images and built a *k*-nearest neighbor model to classify glaucoma and normal cases. They also suggested a GRI based on the pattern features. The range of their GRI was between 2 and 8, and the glaucoma cases were clearly separated from the normal cases. However, the distribution of the GRI reported by Acharya et al. was not continuous. The glaucoma and normal groups were concentrated in specific sections within the entire GRI range. Therefore, it is not sufficient to detect the progression of glaucoma. In particular, the border range between normal and glaucoma cases is difficult to express using the GRI.

As previously stated, several GRIs have been proposed. They have the following disadvantages: (i) Most of them use only fundus images as a resource for GRI. However, ophthalmologists do not diagnose glaucoma based solely on single examination data. The proposed GRIs do not reflect various diagnostic tools for glaucoma. (ii) They are a by-product of glaucoma classification, and their purpose is to predict glaucoma or to obtain auxiliary information for glaucoma prediction. (iii) The risk index should have a range of fully continuous values; however, some GRIs do not. Therefore, they are not suitable as expression/observation tools in monitoring the progression of glaucoma.

In this paper, we propose a new GRI measure named “integrated GRI (I-GRI)”. The goal of the I-GRI is to capture the risk or progression of glaucoma. We used general examination data, including visual field (VF), optical coherence tomography (OCT), and IOP, as sources of I-GRI measurements. Machine-learning schemes, such as feature selection and feature importance, were applied to produce features for I-GRI. The details are described in Section 2 and Section 3. The relevance and usefulness of I-GRI are discussed in Section 4.

## 2. Materials and Methods

To develop the I-GRI, we built a dataset based on the results of the VF, Retinal Nerve Fiber Layer (RNFL) OCT, and IOP tests. After the feature selection process, a machine-learning predictive model using the XGBoost [13,14,15] algorithm with six features (variables) was built to obtain the feature importance values for the selected six features. We also synthesized a new feature using these six features. Finally, we built an I-GRI measure using the seven features. Figure 1 presents an overview of the I-GRI development process. The process is described in detail in the following sections.

### 2.1. Preparation of the Dataset

To build the I-GRI, we collected medical records of patients who underwent RNFL OCT, VF, and IOP examinations at Gyeongsang National University Hospital and Dankook University Hospital in Korea between January 2012 and November 2020. To conduct the study, we required all patients to undergo comprehensive ophthalmological examinations, including slit-lamp biomicroscopy, best-corrected visual acuity (BCVA), autorefraction (KR8800, Topcon, Tokyo, Japan), central corneal thickness (CCT) measurement (Pentacam, Oculus GmbH, Wetzlar, Germany), Goldmann applanation tonometry, dilated fundus examination, and fundus and red-free fundus photography (Canon, Tokyo, Japan) [2]. An automated VF test was conducted using the 30–2 program Swedish interactive threshold algorithm standard on a Humphrey 740 visual field analyzer (Carl Zeiss Meditec, Inc., Dublin, CA, USA). Spectral-domain OCT (SD-OCT) images, obtained using the Spectralis^®^ platform (Heidelberg Engineering GmbH, Heidelberg, Germany), were used to measure the thickness of the peripapillary retinal nerve fiber layer [2].

In total, 868 eyes (of patients) with glaucoma (primary open-angle glaucoma or normal-tension glaucoma) and 1060 eyes (of patients) without glaucoma were included. The inclusion criteria for normal eyes were BCVA ≥ 20/40, no abnormal findings except for insignificant age-related cataract, and normal VF test results according to the Anderson-Patella criteria. Moreover, glaucomatous eyes were defined as follows: (1) BCVA ≥ 20/40; (2) presence of typical glaucomatous changes, such as disc rim narrowing or a notch, and RNFL defects identified by fundus photography and OCT; and (4) glaucomatous VF defect results according to the Anderson–Patella criteria. Table 1 summarizes the characteristics of the participants.

The prepared dataset contained 22 features and a class label (0: normal; 1: glaucoma). Table 2 summarizes the feature list of this dataset.

To build the I-GRI, we needed to collect features that were strongly related to the progression of glaucoma. We applied feature subset selection [16,17,18] to the dataset. We removed 12 less important features using filter methods and obtained six features using the wrapper method. The six final features are listed in Table 3. The final dataset, including the six features, was normalized for use as a reference dataset for calculating the synthesized features.

### 2.2. Building of the Machine-Learning Predictive Model

To obtain the importance of the selected features, we built a machine-learning predictive model, particularly the XGBoost model, and obtained the feature importance from the model. XGBoost is a strong boosting algorithm that has shown a high classification accuracy in many applications. This also supports our calculation method for feature importance. For parameter tuning, we used 10-fold cross validation and obtained the parameter values in Table 4. After parameter tuning, all data were used for the training process because our goal was not to develop the best predictive model but to select the best features for the model.

### 2.3. Calculation of the Feature Importance of the Selected Features

To combine the six feature values into one, the consolidation ratio for each feature is required. The correlation coefficient between the feature and class label is a strong candidate; however, it assumes that the features are independent of each other. However, features interact with each other during the prediction/diagnosis tasks [19,20]. Feature importance captures both the discriminative and interaction powers of a feature. Therefore, feature importance is a proper standard for the consolidation ratio. XGBoost supports feature importance calculations for the features in a model. Table 5 and Figure 2 show the importance of the selected features. RNFL_I is the most important feature for the glaucoma predictive model, and the sum of the importance of the six features is 1.0.

In the proposed I-GRI, the higher the risk index value, the greater the risk/severity of glaucoma. The index ranges from 0 to 1. To achieve these characteristics, each feature value must be normalized in the collected dataset to be between 0 and 1. We applied min-max normalization and excluded outliers. Table 6 lists the minimum and maximum values of the selected features. Among the four features (RNFL_S, RNFL_I, RNFL_T, and IOP), the lower the feature value, the greater the risk/severity of glaucoma. This is contrary to the characteristics of I-GRI. Therefore, we take the reverse values (1—normalized feature values) for these four features.

The performance of the GRI is affected by the quality of the features used to build the index. The addition of higher-quality features should improve the index. In this study, we created a new feature, namely, the “nearest neighbor index (NNI),” from the normalized and partially reversed dataset of six features. The NNI value for a given instance *X* is obtained using the following steps:Take the five nearest instances of *X* from the reference dataset.Investigate the class labels of the five nearest instances.
(2)$$3.\;\mathrm{NNI}=\frac{Number\;of\;instances\;labeled\;\mathrm{as}\;glaucoma}{5}$$

The higher the NNI value, the greater the risk/severity of glaucoma. The range of NNI values is from 0 to 1. Figure 3 shows the distribution of NNI in the normal and glaucoma cases. The boxplot shows that the NNI values in the glaucoma group are significantly different from those in the normal group. This indicates that NNI is a good feature for classifying normal and glaucoma cases.

### 2.4. Building of the I-GRI Measure

The I-GRI is composed of six original features and a synthesized feature. First, the base index value (I-GRI) is calculated using Equation (3) by multiplying the normalized feature value and its importance ratio. Second, the final I-GRI is calculated by combining the I-GRI.base and NNI at a ratio of 8:2. We tested various ratios, such as 9:1, 8:2, and 7:3. The test results showed that 8:2 was the best ratio for I-GRI performance.
I-GRI.base = PSD.normal × 0.27 +MD.normal × 0.14 +RNFL_S.normal × 0.11 +RNFL_I.normal × 0.31 +RNFL_T.normal × 0.1 +IOP.normal × 0.07(3)
I-GRI = I-GRI.base × 0.8 + NNI × 0.2(4)

**Example 1**.*The I-GRI was calculated using the examination data shown in Table 7 as an example*.

Step 1. The examination data were normalized using the minimum and maximum values in Table 6. Table 8 presents the results.

Step 2. The normalized data of MD, RNFL_S, RNFL_I, and RNFL_T were reversed. Table 9 presents the results.

Step 3. The NNI of the data in Table 9 was calculated, using the normalized and partially reversed datasets. The calculation result was 1.0.

Step 4. The I-GRI.base was calculated according to Equation (3). The result was 0.5971.
I-GRI.base = 0.5386 × 0.27 + 0.4667 × 0.14 + 0.6889 × 0.11 +0.7231 × 0.31 + 0.6889 × 0.1 + 0.25 × 0.07= 0.5971

Step 5. The final I-GRI was calculated according to Equation (4). The result was 0.678.
I-GRI = 0.5971 × 0.8 + 1.0 × 0.2= 0.678

## 3. Results

To confirm the validity of the proposed I-GRI, we calculated the I-GRI values for our dataset and analyzed the results. Figure 4 shows the distribution of I-GRI values for the glaucoma and normal groups. As previously described, the higher the I-GRI value, the greater the risk/severity of glaucoma. The overlapped area in Figure 4 is the border section between the glaucoma and normal groups. The shapes of the glaucoma and normal groups show a normal distribution curve with narrow overlapping intervals. This is consistent with the expectations for the risk index. From the results, we calculated the threshold value of the I-GRI to classify glaucoma and normal cases. The threshold was calculated as 0.36. This means that an examination case can be classified as glaucoma if its I-GRI value is greater than 0.36. When we applied this threshold to our dataset, we obtained a misclassification rate of 0.07 (7%). Figure 5 shows a boxplot of I-GRI values for the reference dataset. As can be seen in the figure, there was a clear difference between the mean values of the glaucoma and normal groups. The mean values of the glaucoma and normal groups were 0.607 and 0.249, respectively. The *p*-value from the *t*-test was <2.2 × 10^−16^.

Table 10 and Figure 6 show typical examination data and I-GRI values for the glaucoma, border section, and normal groups. In the radar chart in Figure 6, the gray polygon line indicates the threshold of each feature for classifying glaucoma and normal cases. If an I-GRI polygon covers a gray polygon, the I-GRI value indicates glaucoma.

Figure 7 shows the relationship between the six features and I-GRI. The *Y*-axis expresses the I-GRI value, whereas the *X*-axis expresses the feature value that is normalized and reversed. The value on the right-hand side of the feature name is the correlation coefficient between the feature and I-GRI. RNFL_I, PSD, and MD showed a strong linear relationship with I-GRI, whereas IOP showed a weak linear relationship.

Table 11 summarizes the comparison between the state-of-the-art methods and the proposed method. As we can see, the proposed method supports 0–1 normalization and continuity of the risk index. It uses various resources to capture the characteristics of glaucoma. This demonstrates the advantages of the proposed method.

## 4. Discussion

### 4.1. Effect of the NNI

The NNI is a synthesized feature used to build the proposed I-GRI measure. To demonstrate its effect, we temporarily built the I-GRI, excluding the NNI, and compared the results. As shown in Figure 8, the NNI pulls the I-GRI values for the normal group to the left side and those for the glaucoma group to the right side. As a result, the overlapping area between the glaucoma and normal groups is reduced. Thus, the NNI clearly contributes to the increased performance of the I-GRI.

### 4.2. Comparison between the I-GRI and the MD

Generally, the MD value of the VF indicates the overall severity of visual field loss in patients with glaucoma. Therefore, we can observe the progression of glaucoma using the MD. We used the normal and glaucoma cases from the reference dataset and compared the I-GRI and MD values. Figure 9 shows a scatterplot of the MD and I-GRI values. A linear correlation can be observed between the MD of the glaucoma group and the I-GRI. The correlation coefficient between these variables was 0.803, indicating a strong correlation.

Mills et al. [21] suggested three stages of glaucoma progression based on the MD values. Table 12 summarizes the normal and three stages of glaucoma, and the mean I-GRI values for the stages. We can observe that the mean of the I-GRI values is clearly different for the normal and the three stages of glaucoma and that the *p*-value of the analysis of variance test is <10^−3^. Figure 10 shows the distribution of I-GRI for the normal and three stages of glaucoma. These results imply that the I-GRI reflects the characteristics of MD. Furthermore, the I-GRI can specify the normal and early glaucoma groups. The *p*-value of the *t*-test between the two groups was <10^−3^.

### 4.3. Comparison between Glaucoma and Glaucoma-like Diseases Based on the I-GRI

Several optic nerve diseases must be clinically differentiated from glaucoma. The first is a glaucoma-like optic disc with an increased cup/disc ratio but no RNFL or visual field defects. The second is optic disc atrophy due to diseases other than glaucoma, such as compression optic neuropathy or Leber’s hereditary optic neuropathy. Finally, superior segmental optic hypoplasia is also observed. In this study, glaucoma-like disc was defined similarly to the previous study as an increased cup-to-disc ratio (≥0.6) and pallor, asymmetry of cupping between eyes in a patient with normal IOP, no RNFL defect, and normal VF [22]. The diagnosis of Leber’s hereditary optic neuropathy (LHON) was based on clinical features such as a young patient with progressive central visual loss, color vision defect, optic disc pallor, and central/centro cecal scotoma. Finally, LHON diagnosis was molecularly confirmed at the Laboratory of Dankook University Hospital. Superior segmental optic hypoplasia diagnosis was based on the presence of more than two of the following four symptoms—superior rim thinning, the superior entrance of the central retinal artery, scleral halo, and pale optic disc—combined with nonprogressive VF loss, peripheral VF defects, and IOP less than 21 mmHg. Clinically, ophthalmologists often have difficulty in distinguishing between these diseases, and imaging tests such as magnetic resonance imaging, genetic testing, or serial observation are sometimes required. To help differentiate these diseases from glaucoma, we analyzed the distributions of I-GRI values in the glaucoma and glaucoma-like groups. Table 13 and Figure 11 summarize the results. The I-GRI values of the glaucoma-like group were lower than those of the glaucoma group. This means that the I-GRI can capture the characteristics of glaucoma and glaucoma-like diseases. This demonstrates the effectiveness of the I-GRI. However, the I-GRI cannot be directly used for specifying glaucoma-like diseases because the range of I-GRI values of the glaucoma-like group overlaps with that of the normal group.

## 5. Conclusions

From this discussion, we can confirm that the proposed I-GRI is a reasonable and effective measure for observing the progression of glaucoma. We implemented the proposed I-GRI calculator on the web (http://220.149.235.96:3838/IGRI/, accessed on 10 December 2021). The screen capture of the I-GRI calculator is presented in Appendix A. If an ophthalmologist inputs the patient’s examination data on the site, it presents the I-GRI values and explanation graphs. This can help in identifying the risk of glaucoma and in determining whether a patient has glaucoma.

## Figures and Tables

**Figure 1 diagnostics-12-00734-f001:**
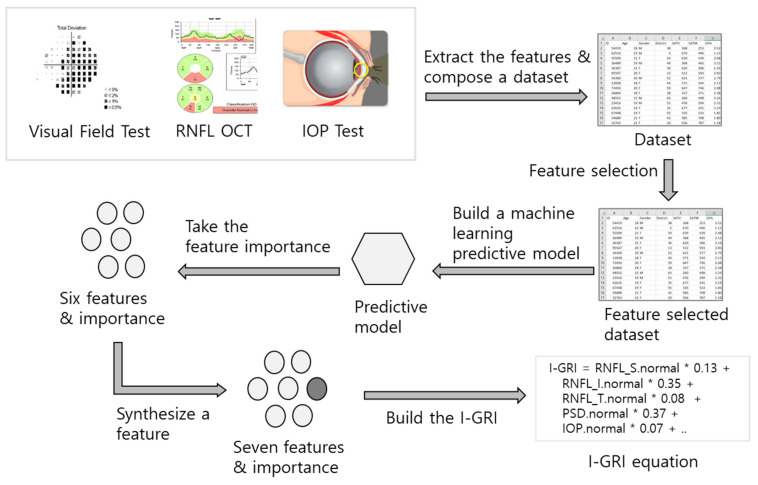
Overview of the I-GRI development process.

**Figure 2 diagnostics-12-00734-f002:**
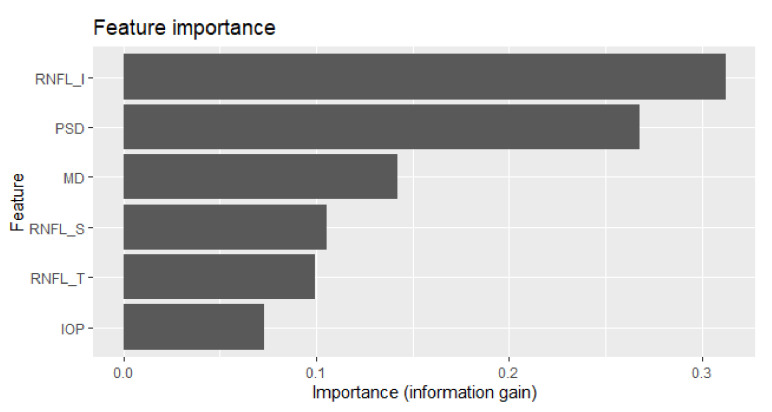
Graph of the feature importance of the selected features.

**Figure 3 diagnostics-12-00734-f003:**
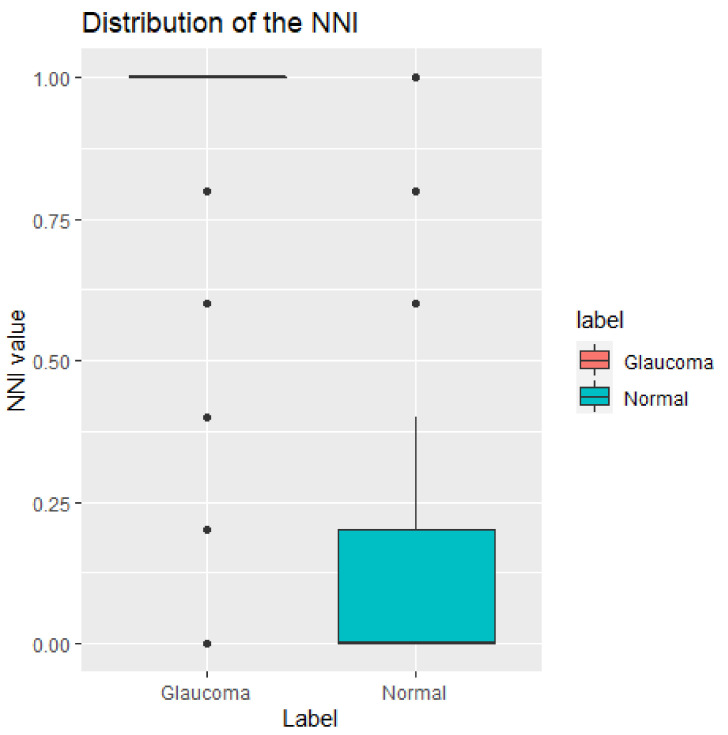
Distribution of the NNI values.

**Figure 4 diagnostics-12-00734-f004:**
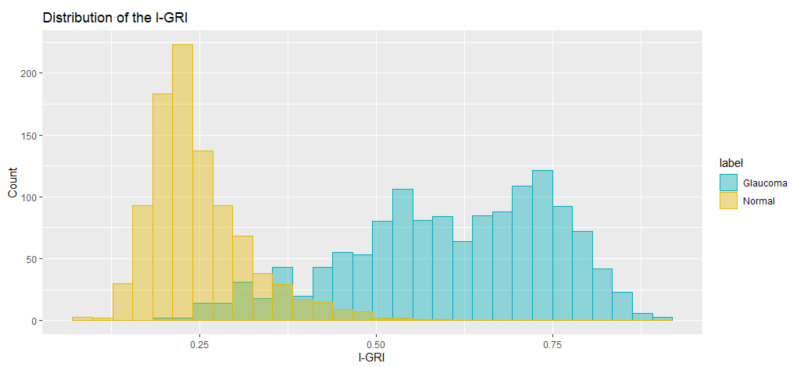
Distribution of the I-GRI values for the reference dataset.

**Figure 5 diagnostics-12-00734-f005:**
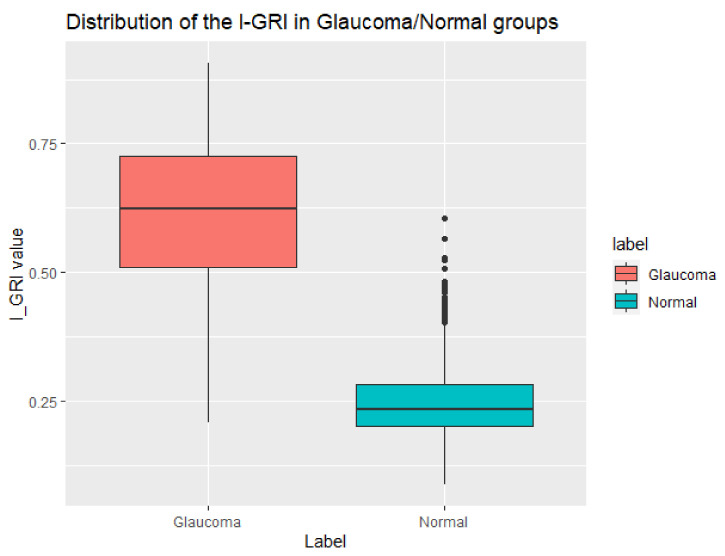
Boxplot of the I-GRI values for the reference dataset.

**Figure 6 diagnostics-12-00734-f006:**
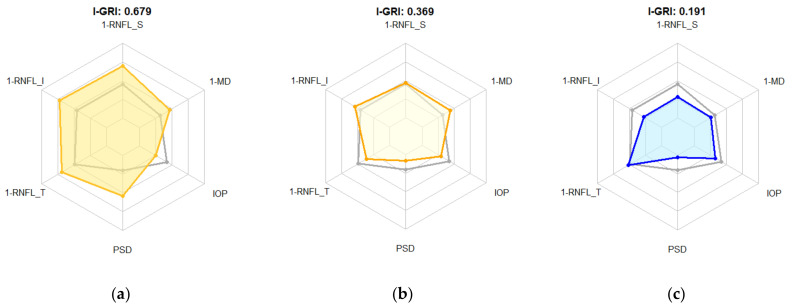
Boxplot of the I-GRI values for the reference dataset. (**a**) Glaucoma; (**b**) border section; (**c**) normal.

**Figure 7 diagnostics-12-00734-f007:**
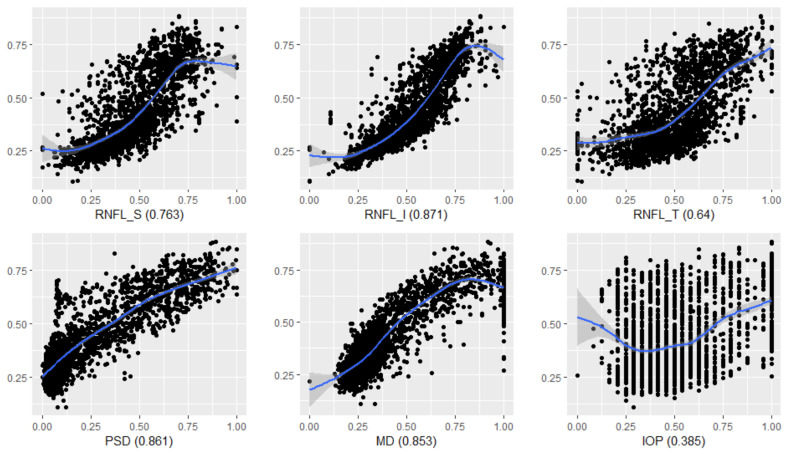
Scatterplots for the six features and I-GRI values.

**Figure 8 diagnostics-12-00734-f008:**
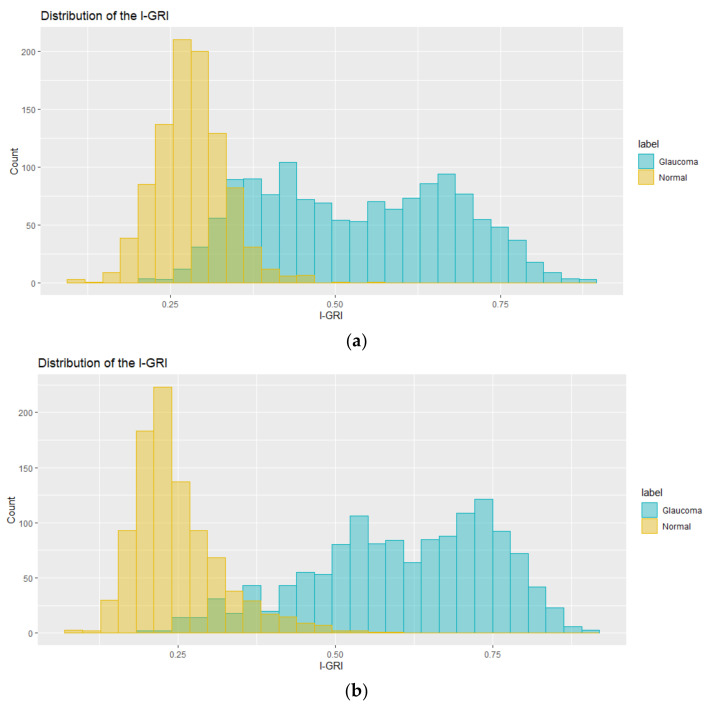
Effect of the NNI on the I-GRI measure. (**a**) Before applying the NNI; (**b**) after applying the NNI.

**Figure 9 diagnostics-12-00734-f009:**
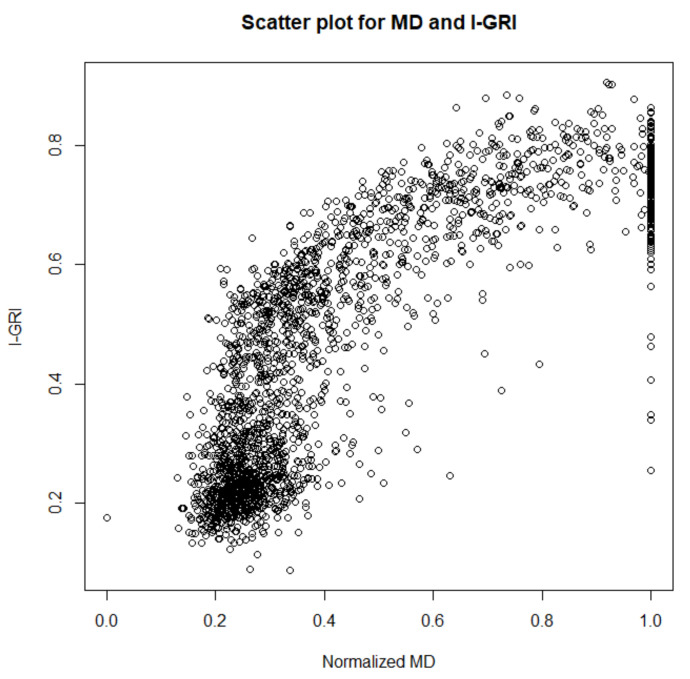
Scatterplot of the MD of the glaucoma group and the I-GRI.

**Figure 10 diagnostics-12-00734-f010:**
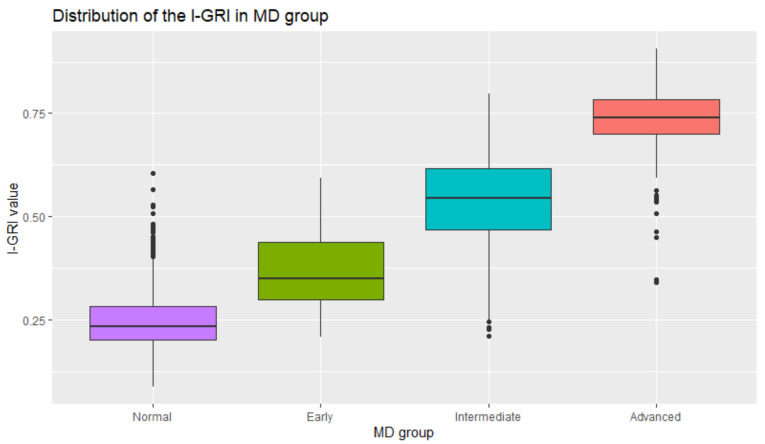
Distribution of the I-GRI values according to the MD groups.

**Figure 11 diagnostics-12-00734-f011:**
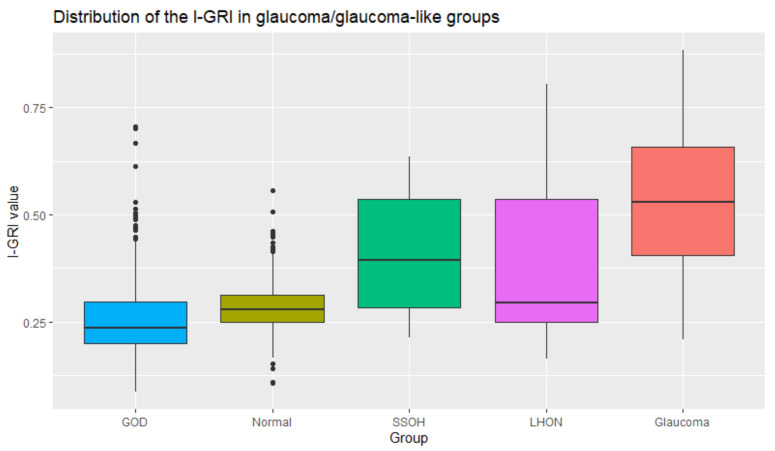
Distribution of the I-GRI values in the glaucoma and glaucoma-like groups.

**Table 1 diagnostics-12-00734-t001:** Characteristics of the participants.

	Normal Group	Glaucoma Group	Total
Number of participants	629	741	1370
Gender (male/female)	518/437	852/497	1370/934
Age (mean ± SD ^1^)	51.1 ± 15.1	59.1 ± 14.1	55.8 ± 15.3
Number of eyes	868	1060	1928
Number of cases	955	1349	2304

^1^ Standard deviation.

**Table 2 diagnostics-12-00734-t002:** Feature list for the prepared dataset.

Feature List
Sex, age, GHT ^1^, VFI ^2^, MD ^3^, PSD ^4^, RNFL ^5^ superior, RNFL nasal, RNFL inferior, RNFL temporal, mean of the RNFL thickness, IOP ^6,^ CCT ^7^, BCVA ^8^, SE ^9^, axial length, neuroretinal rim, cup, disc, mean of the cup/disc ratio, vertical_cup/disc ratio, and CNN2 ^10^ degree

^1^ Glaucoma hemifield test; ^2^ visual field index; ^3^ mean deviation; ^4^ pattern standard deviation, ^5^ retinal nerve fiber layer; ^6^ intraocular pressure; ^7^ central corneal thickness; ^8^ best-corrected visual acuity; ^9^ spherical equivalent; ^10^ convolutional neural network.

**Table 3 diagnostics-12-00734-t003:** List of selected features.

No.	Feature	Abbreviation	Source
1	Pattern standard deviation	PSD	VF ^1^
2	Mean deviation (defect)	MD	VF
3	RNFL superior	RNFL_S	OCT ^2^
4	RNFL inferior	RNFL_I	OCT
5	RNFL temporal	RNFL_T	OCT
6	IOP	IOP	IOP ^3^

^1^ Visual field test; ^2^ optical coherence tomography; ^3^ intraocular pressure.

**Table 4 diagnostics-12-00734-t004:** Parameter values for the XGBoost model.

Parameter Name	Value
booster	“gbtree”
Eta	0.4
max_depth	4
gamma	1
subsample	0.7
objective	“multi:softprob”
eval_metric	“merror”
num_class	2

**Table 5 diagnostics-12-00734-t005:** Feature importance of the selected features.

No.	Feature	Importance
1	PSD	0.27
2	MD	0.14
3	RNFL_S	0.11
4	RNFL_I	0.31
5	RNFL_T	0.10
6	IOP	0.07

**Table 6 diagnostics-12-00734-t006:** Min and max values for the proposed normalization.

No.	Feature	Min	Max
1	PSD	0.95	16.9
2	MD	−24.1	6.39
3	RNFL_S	6	172
4	RNFL_I	0	195
5	RNFL_T	20	110
6	IOP	5	29

**Table 7 diagnostics-12-00734-t007:** Target examination data for the calculation of the I-GRI.

PSD	MD	RNFL_S	RNFL_I	RNFL_T	IOP
9.54	−0.84	56	54	48	11

**Table 8 diagnostics-12-00734-t008:** Results of the min–max normalization for the target examination data.

PSD	MD	RNFL_S	RNFL_I	RNFL_T	IOP
0.5386	0.5333	0.3012	0.2769	0.3111	0.25

**Table 9 diagnostics-12-00734-t009:** Results of reversing four normalized feature values.

PSD	MD	RNFL_S	RNFL_I	RNFL_T	IOP
0.5386	0.4667	0.6889	0.7231	0.6889	0.25

**Table 10 diagnostics-12-00734-t010:** Target examination data for the calculation of the I-GRI.

Group	PSD	MD	RNFL_S	RNFL_I	RNFL_T	IOP	I-GRI
Glaucoma	9.54	−7.84	56	54	48	11	0.679
Border section	2.29	−6.99	94	89	77	12	0.369
Normal	1.43	−1.49	125	140	63	13	0.191

**Table 11 diagnostics-12-00734-t011:** Comparison of state-of-the-art work and the proposed method.

Comparison Point	Bock [8]	Loewen [9]	Mookiah [10]	Acharya [11]	Proposed
0–1 normalization	X	X	X	X	O
Continuity of risk index	O	X	O	O	O
Number of used features	3	3	13	23	6
Resource ^1^	fundus image	IOP, VF, NPM ^2^	fundus image	fundus image	IOP, VF, OCT
Accuracy ^3^	0.80	NA	0.95	0.93	0.93

^1^ Resource to extract features for building risk index; ^2^ number of preoperative medications; ^3^ classification accuracy when the glaucoma risk index is used.

**Table 12 diagnostics-12-00734-t012:** Three stages of glaucoma progression and their range of MD.

Group	MD	Mean (I-GRI)
Normal	–	0.249
Early glaucoma	>−0.5 dB	0.374
Intermediate glaucoma	−5.0 to −12.0 dB	0.539
Advanced glaucoma	<−12 dB	0.737

**Table 13 diagnostics-12-00734-t013:** Three means of the I-GRI for the glaucoma and glaucoma-like groups.

Group	Mean (I-GRI)	*p*-Value *
Glaucoma	0.607	–
GOD ^1^	0.263	<10^−3^
LHON ^2^	0.403	<10^−3^
SSOH ^3^	0.417	0.001

* *t*-test between the glaucoma and other groups; ^1^ glaucoma-like optic disc; ^2^ optic disc atrophy; ^3^ super segmental optic hypoplasia.

## Data Availability

Not applicable.

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
