# Peer review of "Development of the Integrated Glaucoma Risk Index"

_diagnostics, 2022, doi:10.3390/diagnostics12030734_

Round 1

Reviewer 1 Report

In section II, it would be usefull to the lector if you five a little explation of what is the algorithm o heuristics uses by XGBoost  to obtain the 6   selected  features. 

It would be very enriching if you could test your methodology with another dataset with the same characteristics. With this, the endemic characteristic of the data would be removed.

Author Response

Q1) In section II, it would be usefull to the lector if you five a little explation of what is the algorithm heuristics uses by XGBoost to obtain the 6 selected features.

Response:

Six features are already selected in the pre-processing step, and XGboost model is built using the six features. Therefore, the purpose of XGBoost is not feature selection. XGBoost is used to get importance (effect) of the six features.

Q2) It would be very enriching if you could test your methodology with another dataset with the same characteristics. With this, the endemic characteristic of the data would be removed.

Response: We also want to test our methodology with another dataset. But it is difficult to get the dataset because it contains patients’ sensitive information and hospitals do not open the data. Maybe the characteristic of western people is different from eastern people. If somebody has a dataset of western people and applies our methodology, he/she will take proper Glaucoma Risk Factor for western people.

Reviewer 2 Report

In this paper, the authors propose the integrated glaucoma risk index (I-GRI) based on the visual field (VF) test, optical coherence tomography (OCT), and intraocular pressure (IOP) test. They extract important features from the examination data and their importance using a machine learning scheme and integrate them into a single measure using a mathematical equation. The paper is organized well. However, there are some points to be considered. In the abstract, it is expected to present more results of the performance.

  1. The background of the proposed study should be further explained in detail. Some concepts are hard to comprehend without explaining clearly.
  2. Please explore the robustness of the proposed method.
  3. Spiking neural network (SNN) is biologically inspired and low-power-consumption. Please discuss some possibility to use SNN models, including: Efficient Spike-Driven Learning With Dendritic Event-Based Processing; Neuromorphic Context-Dependent Learning Framework With Fault-Tolerant Spike Routing.
  4. Please discuss the potential application of the proposed study in neuromorphic computing, including: CerebelluMorphic; Scalable digital neuromorphic architecture for large-scale biophysically meaningful neural network with multi-compartment neurons; Efficient hardware implementation of the subthalamic nucleus–external globus pallidus oscillation system and its dynamics investigation.
  5. Can you compare more aspects of the proposed study with the state-of-the-art works. Detailed comparison is vital and meaningful to illustrate the novelty and advantages of the proposed work.
  6. Grammar should be further improved.

Author Response

Q1) In the abstract, it is expected to present more results of the performance.

Response: Proposed I-GRI has no quantitative measure to evaluate performance. So, we add following sentence:

In the sanity test using test cases, I-GRI showed a balanced distribution in both glaucoma and normal cases. When we classified glaucoma and normal cases using the I-GRI, we obtained a misclassification rate of 0.07 (7%). 

Q2) The background of the proposed study should be further explained in detail. Some concepts are hard to comprehend without explaining clearly.

Response: The application domain of this work is Ophthalmology. Therefore, some concepts may be unfamiliar to people in other domains. We tried to omit the medical concepts that have little relevance to this paper. If some concepts are need to detail to understand this work, please suggest them to us.

Q3) Please explore the robustness of the proposed method.

Response: We have added the following sentences to section 3:

The shapes of the glaucoma and normal groups (in Figure 4) show a normal distribution curve with narrow overlapping intervals. This is consistent with expectations for the risk index. … (When we applied this threshold to our dataset, we obtained a misclassification rate of 0.07 (7%).)

Q4) Spiking neural network (SNN) is biologically inspired and low-power-consumption. Please discuss some possibility to use SNN models, including: Efficient Spike-Driven Learning With Dendritic Event-Based Processing; Neuromorphic Context-Dependent Learning Framework With Fault-Tolerant Spike Routing.

Response: One of the main ideas of our work is to merge multiple features into a single Glaucoma Risk Factor. In this process, feature importance is used as a ratio of each feature. Unfortunately, the SNN model does not support feature importance. 

Q5) Please discuss the potential application of the proposed study in neuromorphic computing, including: CerebelluMorphic; Scalable digital neuromorphic architecture for large-scale biophysically meaningful neural network with multi-compartment neurons; Efficient hardware implementation of the subthalamic nucleus–external globus pallidus oscillation system and its dynamics investigation.

Response: Now I do not have enough knowledge about neuromorphic computing. After the revision of this paper, I will survey neuromorphic computing to know the possibilities of application of the proposed study.

Q6) Can you compare more aspects of the proposed study with the state-of-the-art works. Detailed comparison is vital and meaningful to illustrate the novelty and advantages of the proposed work.

Response: We have added the following sentences to section 3:

Table 11 summarizes the comparison between the state-of-the-art methods and the proposed method. As we can see, the proposed method supports 0~1 normalization and continuity of the risk index. It uses various resources to contain characteristics of glaucoma. This demonstrates the advantages of the proposed method.

Table 11. Comparison of state-of-the-art works and the proposed method. 

Comparison point

Bock [8]

Loewen [9]

Mookiah [10]

Acharya [11]

Proposed

0~1 normalization

X

X

X

X

O

Continuity of risk index

O

X

O

O

O

# of used features

3

3

13

23

6

Resource1

fundus image

IOP, VF, NPM2

fundus

image

fundus

image

IOP,VF,

OCT

Accuracy

0.80

NA

0.95

0.93

0.93

1 Resource to extract features for building risk index

2 Number of preoperative medications

3 Classification accuracy when glaucoma risk index is used

Q7) Grammar should be further improved.

Response: We got one more proofread from a native speaker, and attached a certificate of proofreading.

Reviewer 3 Report

The authors proposed the integrated glaucoma risk index (I-GRI) based on the visual field test, optical coherence tomography, and intraocular pressure test. This index is also a criterium of glaucoma severity varied between 0 and 1. This criterium is a superposition of six parameters, including intraocular pressure and others measured by visual field tests and optical coherence tomography.

This criterium is rather heuristic because these six parameters present heterogeneous data and can not be considered jointly (from point of view of machine learning).   Another issue from the point of view of machine learning implementation is that experimental methods used for data labeling are also used for collecting data used for prediction model creation. But this approach does not associate with creating a predictive model based on machine learning algorithms! Here, machine learning was used only for the informative features selection and estimation of these features' importance (impact).

In fact, the main result of the paper is the creation of criterium of glaucoma severity varied between 0 and 1 using a combination of six parameters and every of them associated with this disease independently of the rest. This result is quite interesting and helpful for specialists in this field.

   Recommendations:

  1. In section 4.3. “Comparison between glaucoma and glaucoma-like diseases based on the I-GRI” information about groups should be added.
  2. For data presented in Table 11, information about groups should be added
  3. For data presented in Tables 11, 12, the applicability of t-test use should be confirmed.

Author Response

Q1) This criterium is rather heuristic because these six parameters present heterogeneous data and can not be considered jointly (from point of view of machine learning).

Response: In my opinion, most of the scientific ideas including our work are heuristic, but the implementation process should be systematic and have a sound rationale. We adopt machine learning schemes for a systematic approach.

Q2) Another issue from the point of view of machine learning implementation is that experimental methods used for data labeling are also used for collecting data used for prediction model creation. But this approach does not associate with creating a predictive model based on machine learning algorithms! Here, machine learning was used only for the informative features selection and estimation of these features' importance (impact).

Response: As you pointed out, our work is not pure machine learning research. Our goal does not build a good glaucoma prediction model. Machine learning was used only for the feature selection and took features' importance. Because we need to know informative features and merge the ratio of the features for developing single Glaucoma Risk Index.

Q3) In section 4.3. “Comparison between glaucoma and glaucoma-like diseases based on the I-GRI” information about groups should be added.

For data presented in Table 11, information about groups should be added

Response: We have added the following sentences to section 4.3:

In this study glaucoma-like disc was defined similarly to the previous study as an increased cup-to-disc ratio (≥0.6) and pallor, asymmetry of cupping between eyes in a patient with normal IOP, no RNFL defect, and normal VF. [22]. The diagnosis of Leber's hereditary optic neuropathy (LHON) was based on clinical features like a young patient with progressive central visual loss, with color vision defect, optic disc pallor, and central/centro cecal scotoma. Finally, LHON diagnosis was molecularly confirmed at the Laboratory of Dankook University Hospital. Superior segmental optic hypoplasia diagnosis was based on the presence of more than two of the following four symptoms: superior rim thinning, the superior entrance of the central retinal artery, scleral halo, and pale optic disc; combined with nonprogressive VF loss, peripheral VF defects, and IOP less than 21 mmHg.

Q4) For data presented in Tables 11, 12, the applicability of t-test use should be confirmed.

Response: As you can see, t-test is used to compare means of two groups. We test between glaucoma and one of GOD, LHON, SSOH in Tables 11 and 12. So, there is no problem.